# PRISER: Managing Notification in Multiples Devices with Data Privacy Support

**DOI:** 10.3390/s19143098

**Published:** 2019-07-13

**Authors:** Luis Augusto Silva, Valderi Reis Quietinho Leithardt, Carlos O. Rolim, Gabriel Villarrubia González, Cláudio F. R. Geyer, Jorge Sá Silva

**Affiliations:** 1Laboratory of Embedded and Distributed Systems-LEDS, University of Vale do Itajaí, Itajaí-SC 88302-901, Brazil; 2Departamento de Informática, Universidade da Beira Interior, 6200-001 Covilhã, Portugal; 3Instituto de Informática-GPPD, Federal University of Rio Grande do Sul (UFRGS), Porto Alegre 91501-970, Brazil; 4Department of Computer Engineering, University of Coimbra, 3000-370 Coimbra, Portugal; 5Expert Systems and Applications Lab, Faculty of Science, University of Salamanca, Plaza de los Caídos s/n, 37008 Salamanca, Spain

**Keywords:** Notifications Management, data privacy, Internet of things

## Abstract

With the growing number of mobile devices receiving daily notifications, it is necessary to manage the variety of information produced. New smart devices are developed every day with the ability to generate, send, and display messages about their status, data, and information about other devices. Consequently, the number of notifications received by a user is increasing and their tolerance may decrease in a short time. With this, it is necessary to develop a management system and notification controls. In this context, this work proposes a notification and alert management system called PRISER. Its focus is on user profiles and environments, applying data privacy criteria.

## 1. Introduction

Technological evolutions in the urban area have allowed for the integration of sensor networks and devices together with all citizens involved in the daily context. These devices are capable of sensing, processing data and communicating through a network [1]. Although today this term is used more comprehensively, including in healthcare, logistics, security, agriculture, among others, the primary goal remains the same—to make create computers to that capture real-world information without the help of human intervention [2]. Such features, in conjunction with advances in microelectronic systems technologies with new wireless technologies has resulted in the development of smaller devices with considerable processing power, resulting in the Internet of Things (IoT).

IoT applications are seen as a premise with great potential for integrations between the user and the environment. An environment formed by such devices and sensors is called an Intelligent Environment and its primary focus is to bring computation into the physical world and to enhance occupants’ experiences with ordinary activities.

However, with IoT applications, the high heterogeneity of systems, devices and the constraints of imposed resources make it difficult to apply conventional IoT-related security and privacy techniques, thus necessitating specific control and management systems as described by Reference [3].

Such drawback originates from information present in these environments which are shared between applications and platforms, by the premise of achieving device interoperability [4]. The functional capacity of intelligent devices is directly related to the effectiveness of communication, and are used with different technologies, for the most part, wireless. With exponential growth, technologies and their attributions must be appropriately organized and classified to contribute to the scenario in which they are inserted.

In the IoT, several devices are trackable over the Internet, creating threats to personal and private data. Therefore, it is essential that there is a guarantee for data to be addressed to its owner in order to avoid possible risks. All data must be guaranteed not to be used without the user’s permission. A privacy policy is the first step towards a secure data-keeping solution in a smart environment. Whenever another user contacts the data, the privacy policy must be verified before the communication is made [5].

According to Ahlgren et al. [6], IoT is based on three paradigms: Internet-oriented, Sensors and Knowledge. Its primary goal is to ensure the connection between the large volume of devices connected through the Internet, allowing the advancement and development of intelligent environments. Regarding the communication between smart devices, platforms, and the user, there is usually some action from a device and a reaction from the user. For that, notifications and alerts are used.

IoT provides a significant increase in the number of devices, resulting in an increase of the number of notifications driven to users. With increasing the adoption of mobile devices, computers are no longer the only source of interruptions, so there is a need for a management system that addresses multiple devices. In 2016, Mehrotra et al. [7] reported through a survey that a user received 100 notifications on average (per day), every day. This significant influx of notifications also resonates negatively for those who receive the same notifications. With the rise of received notifications, users’ tolerance tends to drop.

In the context of notifications, visual, sound and haptic (vibrations) resources are used to guide the user’s attention with the purpose of delivering instantaneous information [8]. According to these functionalities and with intent to guarantee communication between users and devices present in intelligent environments, it is common to adapt pre-defined requirements and parameters. To do so, according to Reference [3] the following criteria are considered for intelligent and IoT environments: (i) a cost reduction; (ii) improvement of use; (iii) communication related to the user or other devices inserted into the environment; (iv) management and privacy management.

Therefore, this paper proposes a taxonomic model to better organize the parameters and  requirements of the management and control of alerts and notifications for IoT-oriented environments and devices. To provide notification management and integrate applications and  devices, the PRISER (Privacy Services) module has been developed, which has been expanded from the privacy control model for pervasive/ubiquitous environments proposed by Reference [9].

In summary, the contributions made by this paper are twofold: firstly, it presents a taxonomy to serve as guidance to improve privacy in IoT-oriented environments. Secondly, it shows an  architecture of the privacy engine and the conceptual models used for its development.

This paper is structured into seven sections: Section 2 presents the related works; Section 3 is dedicated to the description of the taxonomy; a proposed solution and the test scenario are presented in Section 4; Section 5 presents the prototype and the experimental results and, finally; in Section 6, we present the conclusions and contributions obtained.

## 2. Related Work

According to the researched literature, most contributions are related to privacy control aimed at users and their devices. Initially, most of the work was focused only on computer notifications, and, using external sensors to assist, References [10,11], implemented real-time interruption detection using external sensors and computers, respectively. In these, the used sensors vary from movement, sound, time, location, meeting status (in a meeting or not). and computer activity. In work developed by Reference [12], a DND (Do-Not-Disturb) service was proposed using machine learning techniques to identify the relationship between the current context of a user and the "do not disturb" mode from the device. In this work, the previous user experience was used as a basis, identifying whether the user is available or not.

The exclusive notifications for smartphones using push notifications are addressed in References [13,14,15]. Those proposals convert from mobile and IoT and treat how the system can handle sending them to multiple devices and still treat the communication permeating machine-to-machine (M2M) and human interaction.

Following studies related to notifications, Reference [16] addresses the detection of the moment when the system stops to disturb the user and the adaptation of notifications in real situations. Through a production environment with more than 680,000 real users testing the application for 21 days, in this, we have demonstrated the effectiveness of their system using an Android System [17]. It’s relevance is mainly in the study of the behavior of notifications. Additionally, References [18,19,20] strongly discuss the user’s location to perform an action, or to handle a point of interest (POI). All of these systems focus on detecting breakpoints naturally (a user’s breakpoints, for example,  ending a phone call), detecting breakpoints during an activity or activity limits, and the timing for sending a notification. Regarding the taxonomy, Reference [21] defines the components required for IoT from a high-level perspective, using a taxonomic model for such a task. Literature such as References [22,23] and current works such as References [24,25,26] use specific taxonomies to classify components and sensors with the aim of contributing to the development of new sensors for IoT.

The approaches of References [16,27,28] make it clear that a decision method should be used to evaluate the user’s moment of interruption. On the other hand, the approach of Reference [29] clarifies regarding oncoming devices used to send notifications. Reference [30,31] points out the use of a multi-device environment to improve delivery processes, which we intend to  apply in this research.

The work of Reference [32] is preliminary and attempts only to predict which device should be selected to deliver the notification, however, the set of notification data used to train the algorithms and evaluate the system result is partially synthetic and assumes that the data available for the notification is explicit. In contrast, the management system proposed by Reference [33], attempts to predict the most opportune moment for the notification to be delivered to the user using a set of data obtained from real users. Only the abstract derivation is used by the manager to predict delivery time. This approach is used in the same way in Reference [34], as they utilize this to protect the privacy of the user’s location, which had its data collected, with information about the implementation of the notification management solutions and location services. In Reference [35] the multi-device approach and the use of an application to collect notifications demonstrate efficiency, however, the application performance is not measured.

We analyzed information about the implementation of notification management solutions and  location services. The information obtained in the analysis of the selected works are presented in the Table 1, the items that the proposal needs or uses in the work content are marked with a text, if the solution does not address or does not use, the item appears blank. The table is organized into seven columns as described below:Reference;Solution Name and Year;If the implementation makes use of the user’s location;Whether the implementation determines privacy preferences or implements actions to maintain private data;Compares whether the notification solution makes use of multiple devices, being more than one mobile device or another smart device;Informs whether the proposed solution is applied directly to the user’s mobile device through an application;Informs whether the solution uses human participation to verify the relevance and manual adjustments in the context of notification management.

Although related work addresses location and notification management, many of these studies do not address the privacy features of the user’s environment. Therefore, the present work proposes a module of notifications, to make the environment informative and dynamic to control privacy parameters. The user profile, location, type of environment, criteria, priority and user preferences will be considered to define the notifications and/or alerts based on individual control and management.

## 3. Taxonomy

According to Reference [36], the term "taxonomy" itself comes from the Greek and is a compound of taxis, meaning order, and nomos,  meaning science. Taxonomies are obtained by splitting a general usually complex concept, idea, or artifact in concepts, the classes, which are progressively more and more specific. All members in a class are marked by the same subset of shared features. Taxonomies allow for a greater degree of precision in the classification process and support known-item seeking strategies, when users already know what they are looking for, very well.

In another sense, the terms defined by a taxonomy are structuring, strategic and central elements used to name, classify, and organize entities into groups that share similar characteristics. The taxonomy is developed from keywords and concepts so that the contents are categorized [22]. The concept of taxonomy becomes advantageous in a large volume of information, and an example is the diversity of IoT equipment growing exponentially. In such a way, the users acquire an essential role both in the production, as well as in the categorization and use of generated information [21].

We present a taxonomic model that defines the components necessary to associate IoT, assist the Notification Management System (NMS) and privacy [24]. However, from a high-level perspective, each component of this taxonomy was based on a series of related works that helped us identify the needs of the NMS. The works used are represented by Table 2.

In this section, we discuss some enabling technologies, which compose the proposed taxonomic model. The taxonomy serves to classify rules and parameters, resulting in a better understanding of the functionality [51]. Our taxonomy is based on four main parameters: (i) communication technologies; (ii) message transmission technology; (iii) privacy and (iv) criteria, with open possibility of expansion of the sub-parameters. Each component of the model is duly described in the following subsections and the graphic representation in Figure 1.

### 3.1. Communication Protocols

Connectivity requirements are essential components and careful attention must be paid to using alternatives in case of failure. In particular, smartphones are considered fundamental in creating the so-called opportunistic paradigm of IoT [38], merging users (carrying mobile devices) and smart objects (smart-things). Thus, the present communication technologies are basically composed of the same ones used in the IoT ecosystem and listed as main by Reference [21]. Having as a requirement that smartphones should always be connected, they are shipped with various communication interfaces (e.g., Wi-Fi, NFC, Bluetooth) [39]. Regarding power consumption [43] demonstrates that delaying the delivery of notifications can generate power savings on mobile devices.

Wi-Fi: Widely-used wireless network technology with high bandwidth. Wi-Fi is still used to estimate location in relation to your access point in location-based-systems [37];Mobile Data (4G/5G): Mobile data 4G and in the future 5G;Bluetooth: It supports a lot of connected equipment, at a low range with an efficient energy management;NFC: Used for payments and short-distance communication, used for authentication in environments being an alternative to the use of traditional keys;RFID: It is used for short and long-range communications. Also for authenticating the location of a  device or user and sending notifications to users in intelligent environments.

### 3.2. Message Transmission Technologies

Cloud-based services for sending text messages (SMS), push notifications to mobile devices, or in recent cases phone calls. These components, including the parameters, work as a single system to give the user complete control over their notifications and it must ensure interoperability between devices in intelligent environments. According to the comparison of Reference [40] the WhatsApp instant messaging application’s popularity over conventional SMS, it concluded that SMS is considered more formal, trustworthy and better for privacy. WhatsApp, on the other hand, is considered informal and is used in a more conversational style. Yet research reports a concern amoung users with the increasing number of notifications and interruptions caused by messaging applications. Placing the phone in silent mode helps address messaging overload, but the authors say that users who use the WhatsApp application and SMS for business-related communications do not mute their phones.

SMS: Short Message Service (SMS) is a service available in digital cellular phones that allows the sending of short messages (up to 160 characters) between these devices and between other ones that use the GSM interface;Push Notifications: Push notifications that appear in the foreground on users’ devices at random times, causing outages. Generally generated from previously-installed applications [16];Instant Messaging: Messages through applications, with the evolution of device programming, the number of applications for this purpose grows;Call: Phone-to-phone calls.

### 3.3. Privacy

Privacy is the goal of specific studies such as Reference [50], a legal expert, has proposed a taxonomy of privacy breaches. It focuses on civil liability law, unlike Reference [24], which addresses a taxonomy focused on privacy-enhancing technologies. Privacy requirements define user permissions through environments and/or devices. With the attributes of the environment, it is possible to verify and manage parameters that compose it. In addition to resources for users, according to their availability and location, as well as the tracking system found in Reference [51].

User: Defining the user as the center of custom alerts notification services, needing to have location contexts and well-defined individual information such as their preferences;Environment: The environment that the user accesses or requests access to, controlling privileges parameters in comparison to the profile;Device: Refers to the device that the user utilizes, from an M2M communication and using the connectivity standards defined in Section 3.1.

### 3.4. Criteria

The Criteria are defined by rules for access, use, sharing, location, etc. Rules can be added, changed, modified and/or replaced according to the environment and with established rules. The settings are handled individually by other modules that have features and controls such as Reference [33], criteria are used as input for processing, and finally setting the timing of sending the notification. The operating settings are predefined for each environment and may have variations and different criteria defined according to the access time.

Priority: The priority is directly related to the notification priority. An example of a high-priority notification would be a climate alert, which should reach the attention of the user, warning of a possible phenomenon;Time To Life (TTL): Time that a notification becomes inefficient since there is no time for delivery of it. A notification with a short lifetime and not sent with the proper priority loses its sense of notification. Likewise, it prevents the sending devices from carrying old messages for a long time [38];User Profile: Defines the limitations for each user. Reference [9] defined that for each environment a user profile type is assigned according to its location and attribution to the user. This environment is according to shift, day, week, month related;Environment Type: The parameter definitions used were based on environments with public, private and restricted location characteristics.

## 4. Priser

The proposed solution is based on UbiPri [9], privacy and control middleware that differs from other existing solutions by providing general privacy management and a control model. The middleware UbiPri has a structure divided into components, each with its specific characteristics, and is used as needed. One of its components refers to services, and this is called PRISER, which is responsible for managing notifications.

According to the researched literature, the privacy taxonomy is defined by necessary components and according to the use in ubiquitous environments. In this study, it is used as a set of rules necessary to control and manage data privacy. The proposed solution uses the criteria for managing the delivery of messages dynamically, according to the situation and location-related privacy requirements.

The main contribution of this work is the control and management of notifications based on  the taxonomic definition presented in section III, taking into account the privacy of the user concerning the environment. The role of taxonomy is to define and share its rules and parameters with the Notification Management System (NMS). This utilizes the user’s location based on their preferences and defines the role for sending notifications. The characteristics of the medium to send messages are context-aware like in Reference [33]. This process of sending notifications are based on criteria, such as type of environment, user profile, time for delivery and priority. Demystifying the types of environments, these can be classified as public, private, restricted and personalized [9].

Based on the criteria for the management of alert sending it is possible to model the sending of  messages dynamically in various embodiments. As a result, the severity or criticality level can be assigned to user notification based on system standards (e.g., severity or criticality levels associated with specific currentl events that may require user attention). Alternatively even custom criteria (e.g., users who can modify system defaults, create custom events associated with severity or user-defined importance levels, etc.). That information along with published messages are broadcast to subscribers (members) of a topic group over dynamically-built spanning trees rooted at the publisherin which it will be consumed from a broker. In this case acting as an intermediary, responsible for storing and queuing events to be notified, as modeled in Figure 2.

In a similar vein, it should still be possible to assign a lifetime value for notification (TTL). In the case of a device without Internet access, in a given environment, it should be possible to send medium and high-priority notifications in another way, such as t hrough an SMS or, in extreme cases, a call. Therefore, NMS uses the publisher/subscriber model as in Reference [53].

### 4.1. Notification Collector Module

The proposed system receives the notifications from different external sources and stores them in the first module (Collector). The information present in the notifications regarding the criteria and the privacy are the exclusive domain of the decision module. The collection has a buffer function, temporarily storing the data while being moved and processed by the decision module.

A mobile device application was developed to gather and record notifications executed in  background. All requirements and running jobs in the background are confident and can be run in almost all android versions. After granting user’s rights, that service is executed permanently in background and receive a callback when a notification is added or removed in the system. Newer Android versions received substantial API improvements, per example, providing information if a notification was removed by the user or by itself. That API is available since Android 4.3 which runs in 96.40% of all Android smartphones [17]. That service provides which applications have set and removed notifications, like text content, priority levels, vibration patterns and added attributes of the  likes. Its application is represented with two screen-shots in Figure 3.

### 4.2. Decision Module

The notifications are then sent to the second module, whose primary function is to make decisions, receiving information related to the privacy of the environment, device, or even the user. It is also receiving criteria information in the context of the user (e.g., location, status, current activity) as well as information relevant to the notification such as lifetime. The criteria has important functions in the NMS and a flow chart represent this in Figure 4. The information is used to choose the best devices and the best forms (e.g., vibration, sound or light) to display the received notifications.

### 4.3. Dispatcher Module

Finally, the dispatcher adapts the notifications to the chosen target devices and sends them. When handling notifications addressing only one device, it causes certain problems. The first point is that the user should always be charging, or close to the device. The second point refers to the connectivity, the device that the user utilizes may become disconnected, or even without a battery. The dispatcher module architecture is a compound of multiple devices as shown by Figure 5. It could be implemented in a distributed manner, overcoming any eventual bottleneck or single point of failure.

### 4.4. Scenarios

To exemplify the use of the proposed PRISER system, this section aims to present scenarios of  use, presenting real situations in which the person could benefit and which are already addressed in the researched literature. Thus, the integration of uses and devices is seen as a contribution.

That is, in order to display notifications not only on the user’s smartphone but in the environment. The use of the environment to communicate information to the user has already been explored according to the researched literature. For example, using the ambient light information system in an office [54], an environment to view text messages [55], or a smart TV as the central notification view [56]. All of these projects, however, focus primarily on displaying notifications similar to those generated on smartphones and bringing them to the environment. In contrast, the present work focuses on integrating the notifications that are generated in the environment itself. We present below potential areas to use the management system:Access control In an access control scenario, the smart home system recognizes visitors at the doorway and sends a notification to the user. For this, it should be possible to adapt the system to the  environment used. An immediate reaction is necessary, otherwise, visitors may assume that no one is at home. In an attempt to gain unauthorized access, the system user may request to be  notified of any attempted improper entries;Environment control In the case of window control of an environment, it begins to rain while a window is open. To prevent damage to the interior of the residence, the system informs the user throught a notification. In this case, the urgency level is quite high and the user should react in the next few minutes;Medical Area The user must apply a medication every night. The intelligent house system detects from a schedule that it must be applied and informs the user through a notification. In this case, the user should react soon [57];IoT A clever vase detects that a plant needs to be watered and sends a notification. The level of  urgency of this notification is quite low and you should react to it in the next few hours.

While in scenarios in which the user is available to receive notifications, it may be useful to be  notified without a level of filtering without causing a disturbance, other scenarios may require that such warnings be automatically discarded (e.g., during sleep), depending on the level of urgency. According to Reference [19], it should be possible for the user to create custom notifications for different situations. The customizations can differ from user to user, and can also have variations such as time, day, week, and still working days [20]. In this way, we explore how the notification delivery device and the location where the notification is displayed can be linked and how notifications can be viewed to convey content implicitly.

## 5. Prototype and Experimental Results

To evaluate the best approach to collect notifications applicable to notification management, a preliminary version of the responsible module was prototyped and tested using a reduced data set. A commercial application namely Pushbullet was used to recover lost notifications and help us test the module responsible for collecting notifications. Such application ensures that a new notification is not missed by collecting and storing lost phone calls and notifications. The first tests with the commercial application were not satisfactory because they could not integrate with the proposed system.

Given this, the PRISER Notification Collector application was implemented and used for the initial experiments. In these experiments, two smartphone devices with an Android operating system were used. The application proposes that the user also can send text messages and receive texts on their device, and respond to messages from various messaging applications between all devices connected to the application. The application remains in the background and collects various system notifications, messages, and even links. The events usually appear in the notification panel, as shown in Figure 6.

The NMS uses the information originated through the device’s operating system. These  notifications are logged with the date and time of the notification activity. The application also registers the following attributes:Name of the event that originated the notification (email account, smart-thing (e.g., washing machine, SmartTV), WhatsApp, and individual users or groups);Event status regarding which type of notification is being sent (screen activated, screen off, and screen unlock);Action performed by the user (notification received, notification removed or still responded);Message content;Time and event data.

Notifications are stored in devices’ memory and can be navigated by the device administrator. A JSON object composed of all notification information is obtained according to the items mentioned above. This can be seen in the image as presented in Figure 7.

### 5.1. Collect and Share Method

The notifications collected in the Android device are shared with the decision module using the publish/subscribe option that implements a MQTT protocol. The websocket unit provides the communication layer between the client-side and the server-side MQTT combination. The reaper unit receives heartbeat events from device workers and the component for connect the devices is called triproxy because it deals with three endpoints instead of the usual two. They can have more than one instance running simultaneously, and then give a comma-separated list to the provider.

The unit below uses a provider in the dispatcher module unit for connects to ADB (Android  Debug Bridge) and to start worker processes for each device. It then sends and receives commands from the processor. Its purpose is to send and receive requests from the app units, and distribute them across the processor units. The architecture explained above of the uses of Message Queue are showed in Figure 8.

The application initially obtains the notifications. This stage of the process integrates the collector module, which makes the sharing with the decision module, the second stage of this work. The onNotificationPosted method plays an important role in NMS, depending on the other methods to execute. This method is showed in Figure 9.

We used commands to access resource measures, executed through the command line, through the debugging tool ADB, made available by the Android operating system itself. The list of device notifications can be retrieved using the dumpsys notification command.

To verify memory usage we used dumpsys meminfo command, for CPU we used dumpsys cpuinfo command, both the commands must use the name of the application as a parameter. In order to check the size of the database, the command ls -l [path] and /stat/[pid]/stat and /proc/stat were also executed. However, these were not required for the analysis, since the dumpsys tool already displays usage statistics.

### 5.2. Collect Notifications in Multiple Applications Scenario

The comparison experiment was done on a computer, used to communicate with devices, evaluating the behavior of all devices together. Therefore, four experiments were performed in a 24-h interval: two with the collector component connected and two with it turned off. The goal is to evaluate the number of notifications received in that interval, memory usage, CPU and amount of storage done by the database. During the test period, devices received notifications about applications, text messages (SMS), messaging applications, operating system notifications, and even missed call reminders.

The Android operating system by default already implements a notification manager, so by  running the above command dumpsys notification, we get a list consisting of every current state of the manager, including counters. Figure 10 represents the chart of generated files. It is important to highlight the numEnqueuedByApp attribute, which is responsible for the number of notifications already queued by applications. demonstrating the number of interruptions caused to them, by applications in the smartphone.

When using notification by application, we only noticed a large accumulation of notifications only of the operating system. This resulted in a new comparison of System Notifications × Notifications of an application showed in Figure 11.

On the one hand, we observed that the higher the number of notifications of applications, the greater the number of notifications of the operating system, and also the latency in the transfer to the decision module. The system can be configured to ignore or filter only relevant messages from the operating system. Such as shutdown routines and updates or battery alerts. Therefore, the collection system proved effective in its main function.

### 5.3. Devices

The devices used for testing were selected because they are commonly-used models, with processing and memory between the averages of the most-used devices. Additionally, the computer used for testing does not have a high processing power simulating also a common user. The devices used for tests are:II Motorola Moto X Play (XT1563)

A computer with the following characteristics was used for the tests:CPU: Intel^®^ Core i5 2.7 GHzRAM: 8 GB (1600 MHz)

The results obtained demonstrated the concepts presented in the course of the work, were based on the taxonomy and continued as rules and definitions according to a researched literature. The evidence is the contributions of this paper, as follows.

## 6. Conclusions and Future Works

Throughout this work, it was possible to identify the importance of using the mentioned criteria, taking into account the user hierarchy, privacy criteria according to the environment and the hierarchy assigned, being possible to define the type of alert.

As a result, we proposed the notification management system with a focus on user privacy. This way, it contributed to developing an application with the treatment of different types of notifications. Additionally, it was possible to guarantee the sending and/or receiving of relevant messages according to previously-defined rules. The related works present limitations and did not define an architecture and/or model with a contribution to work with multiple devices. Therefore, this work also contributes to relating IoT requirements and definitions.

We present an architecture divided into three main modules to manage the notifications received. The present solution made it possible to decide who should receive the notification of receipt, on which device, moment and in which mode (vibration, sound, light). The results allowed us to validate the prototype developed based on the privacy rules. Through these results it was also possible to observe that to solve the problem related to the availability of notifications, these must be treated individually according to their characteristics as they presented our results.

For future work, we point to the development of other methods to employ machine learning algorithms to infer in the decision module. Another line of research is related to the trigger module; we are also testing variations between the MQTT, CoAP and OSGP protocols in order to treat different messages in different devices and types of messages. For the execution of large-scale alerts, the first simulations pointed to the need for code optimization. We also started to implement the security of messages sent and run tests utilizing encryption using protocols mentioned earlier.

## Figures and Tables

**Figure 1 sensors-19-03098-f001:**
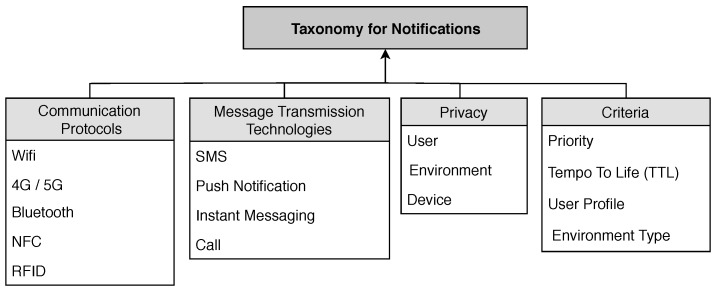
Taxonomy for Notifications Management System.

**Figure 2 sensors-19-03098-f002:**
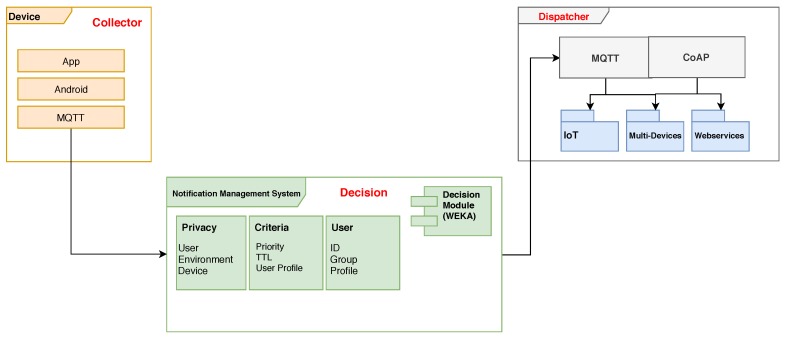
Notifications Management System Architecture.

**Figure 3 sensors-19-03098-f003:**
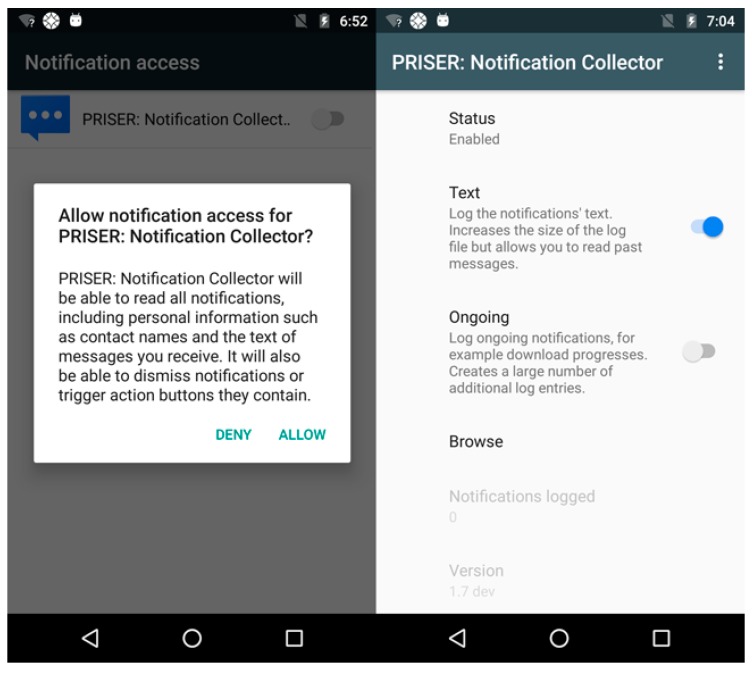
Notification Log Application on Android.

**Figure 4 sensors-19-03098-f004:**
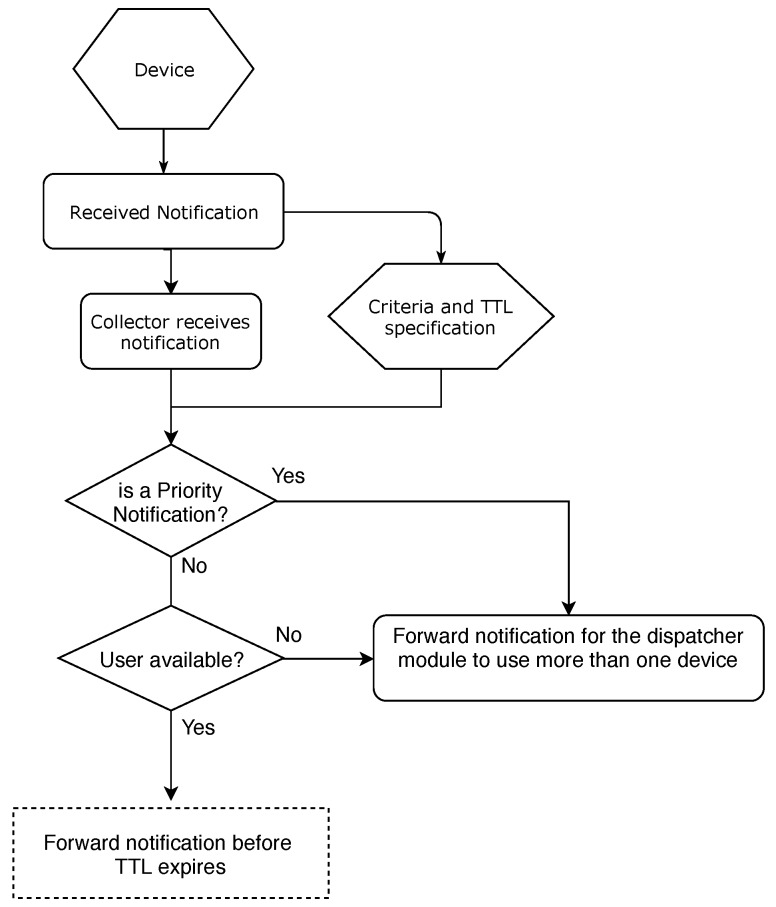
Flowchart NMS.

**Figure 5 sensors-19-03098-f005:**
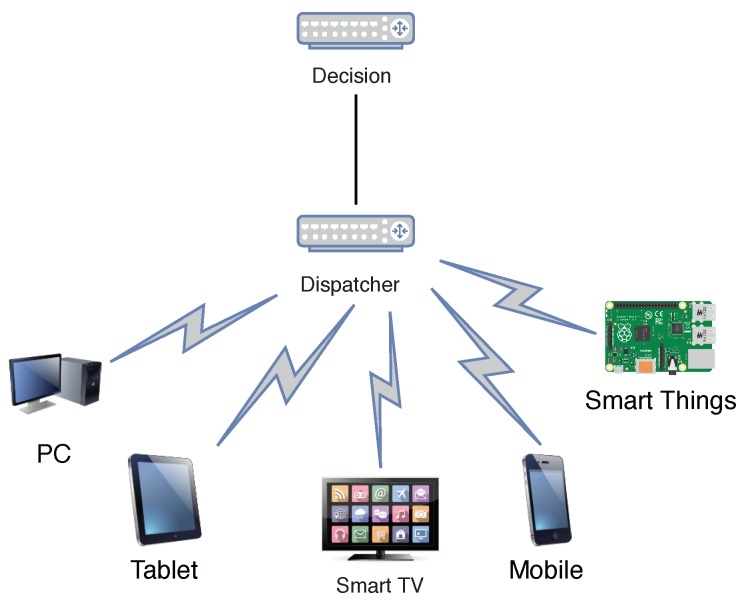
Dispatcher to multi-device.

**Figure 6 sensors-19-03098-f006:**
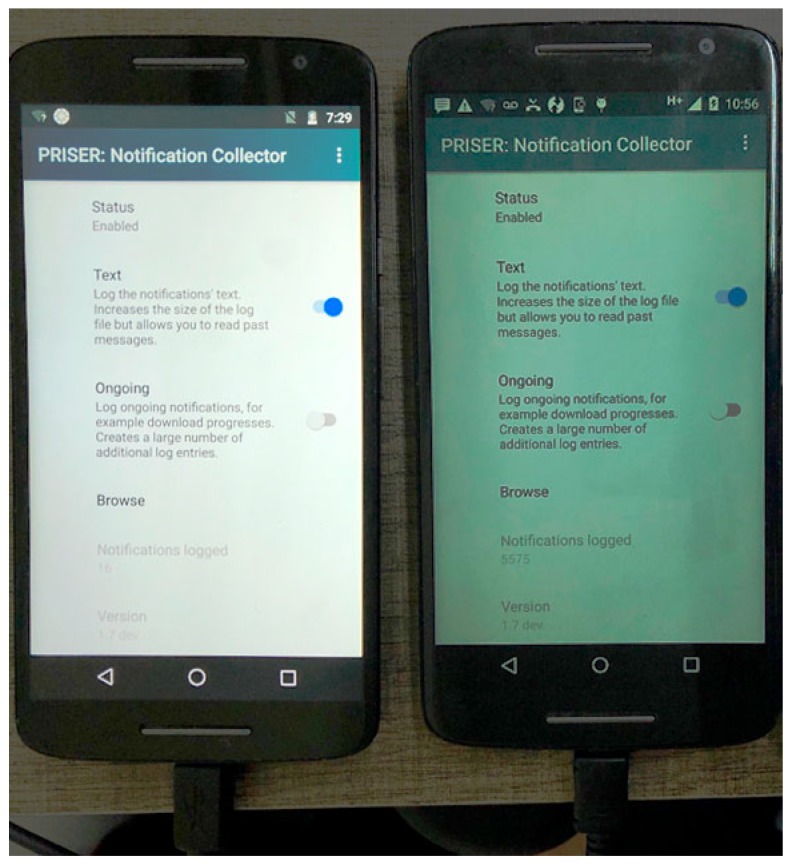
Devices used for primary tests.

**Figure 7 sensors-19-03098-f007:**
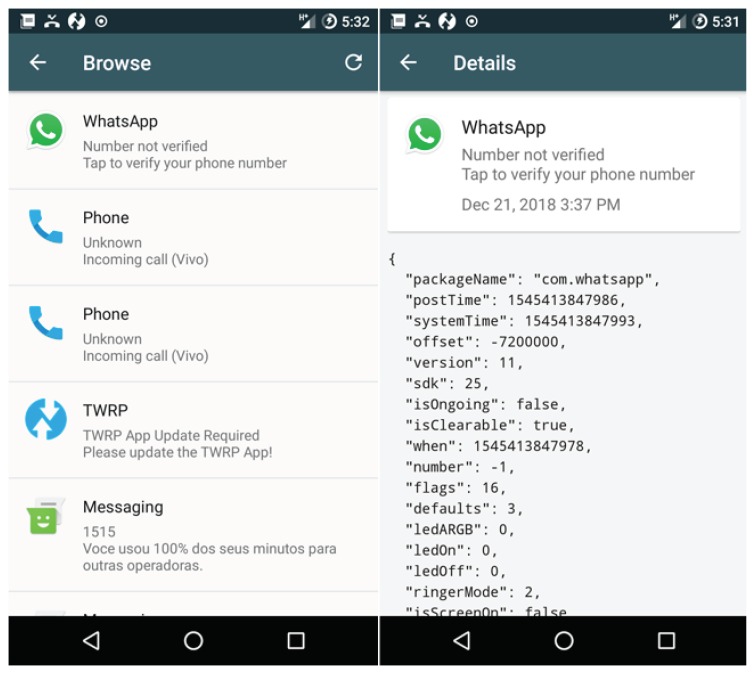
List notifications and details for only one.

**Figure 8 sensors-19-03098-f008:**
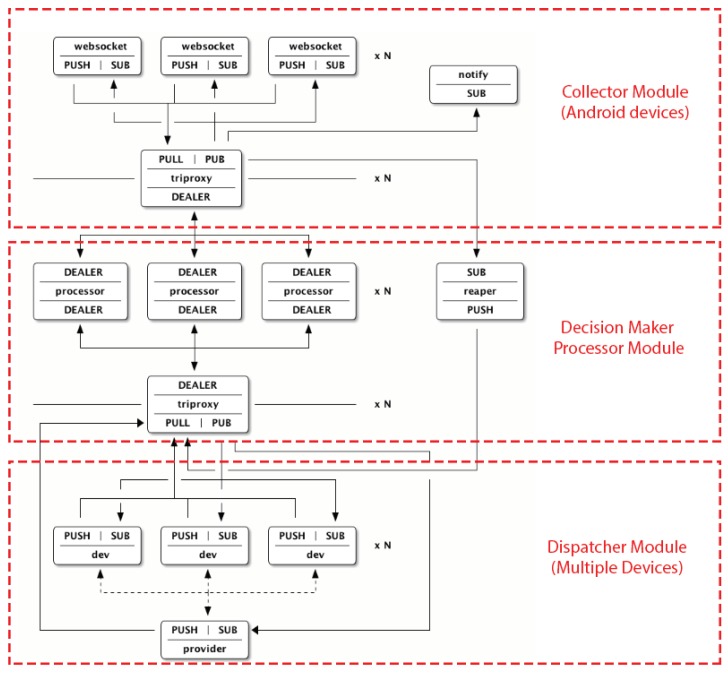
MQTT topology with the brokers and dealers.

**Figure 9 sensors-19-03098-f009:**
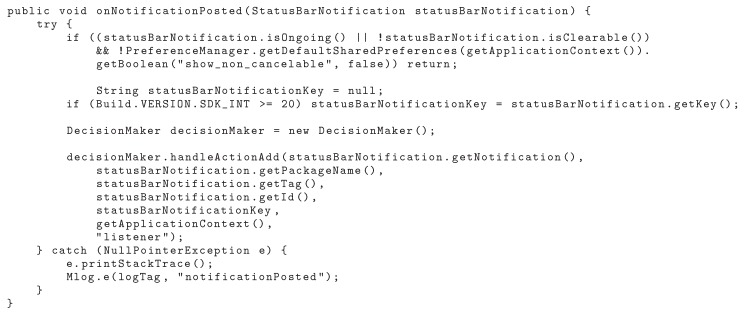
Client Application: Method to obtain the notifications and their attributes.

**Figure 10 sensors-19-03098-f010:**
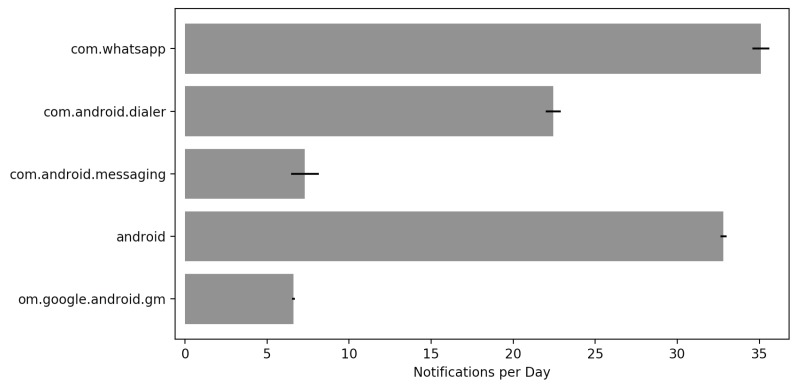
Notifications per application.

**Figure 11 sensors-19-03098-f011:**
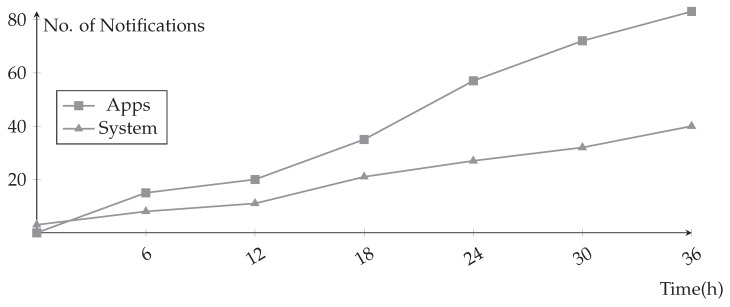
Notification by System × Notification by App per hour.

**Table 1 sensors-19-03098-t001:** Related Work.

Work	Solution	User Location	Privacy	Multiple Devices	User App	Human Intervention
[27]	Attelia (2014)	X				X
[28]	Message Monitor (2014)	X	X		X	X
[29]	Face-to-Face (2014)	X			X	X
[30]	Desktop Notifications (2014)	X	X		X	X
[31]	Intelligent Push (2015)		X		X	
[32]	Notification Collector (2015)	X		X		X
[33]	NAbsMobile (2016)	X	X		X	X
[34]	No name (2017)	X		X	X	
[19]	Smartnotify (2018)	X			X	
[35]	Notification Log (2018)	X		X	X	X
This Work	PRISER (2019)	X	X	X	X	

**Table 2 sensors-19-03098-t002:** Works used to define the components of the taxonomy.

Component	Work
Communication	[21,37,38,39]
Notification	[8,13,16,25,40,41,42,43,44,45,46]
User-location	[29,34,47,48,49]
Privacy	[24,50,51,52]
Criteria	[12,19,33]

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
