# Peer review of "PRISER: Managing Notification in Multiples Devices with Data Privacy Support"

_sensors, 2019, doi:10.3390/s19143098_

Round 1

Reviewer 1 Report

The authors address a notification system in several devices with data privacy support. Below I am presenting m y comments.

1/ Enhance the Abstract. Present the problem and how related work does not solve the problem. Present your contribution.

2/In the introduction, your motivation is not clear. What is the current problem in the literature? Could you provide a figure regarding the problem and another figure with your solution in this part of the text?

3/ Privacy is a well-known topic in security and distributed systems. It is not clear your novelty really. Are you proposing new algorithms? What is the novelty in terms of architecture?

4/ The introduction must be reorganized: theme, sub-theme, problem, related work does not solve the problem, your solution (model) with scientific contribution and method, text organization.

5/ How did you select related work?

6/  In the related work, your works are not new; they are old. Why? How about related work from 2019?

7/ Your article is a mix between survey and proposal. It is strange to me.  How did you assembly the taxonomy? That is the literature corpus? And sources? And key words? And exclusion criteria?

8/ where is the novelty? What section?

9/ how are you addressing scalability?

10/ You are mixing model and technologies. Please, detach your novel ideas from technologies. For example, subsection 4.1 present Android and libraries. But It is possible to work with IoS, right?

11/ You're decision module is very simple. Are you using some robust algorithm?

12/ In conclusion, present how did you achieve the contributions. And how about contributions for the society?

13./ What are the limitations of the work?

14/ You have 53 references, but only one from 2019. Why? Are you up-to-date, really?

Author Response

Dear Reviewer 1,

We thank you immensely for your criticisms, suggestions, and requests about our work. His notes, in addition to contributing to the improvement of this work, will also serve as a basis for our next steps in future research. We agree with your comments and with that, we have improved our article and worked out the answers to resolve all your questions, as follows.

Reviewer 2 Report

The authors introduce a modern notification service for IoT services spanning mobile phones, sensors, smart tv and more. 

I find the motivation behind this paper very appealing and although this can be characterised as a more software engineering problem than research, there is scientific merit to developing such a service.

However, there are a few things that can be done to significantly improve the quality of the manuscript:

- The authors mention as a prominent problem the number of notifications that the user receives. However, there is no evaluation on the success of the framework to actually reduce the number and the accuracy of the method.

- I find that although the authors tackle the problem from a privacy perspective, the authors should at least mention in the taxonomy (and maybe as a future work idea) that notifications are a prominent reason for significant energy consumption and thus less battery life.  I'd suggest the authors look at works that attempt to reduce the processing and dissemination rate of IoT services [1][2] and also studies that show this specific impact to android and iOS.

- The authors should state if the app is actually offered or not.

[1] "Low-Cost Adaptive Monitoring Techniques for the Internet of Things", D. Trihinas and G. Pallis and M. D. Dikaiakos, IEEE Transactions on Services Computing, 2018

[2] "Adaptive Monitoring Dissemination for the Internet of Things", Demetris Trihinas and George Pallis and Marios Dikaiakos, IEEE INFOCOM 2017, 2017.

Author Response

We really appreciate your revision, 

Thanks for the suggestions.

We add this article "Adaptive Monitoring Dissemination for the Internet of Things", Demetris Trihinas and George Pallis and Marios Dikaiakos, IEEE INFOCOM 2017, 2017. in our references, because is aligned with the main problem. 

The application prototype was made for Android devices, at the end of this survey will be made available to the public. One of the lines of research studies how to adapt to iOS.

Reviewer 3 Report

The authors propose a notification and alert management system called PRISER. Its focus on user profiles and environments, applying data privacy criteria.

The paper is well written and is scientifically sound. The pictures have good quality and the tables are well explained throughout the paper.

The Related Work section also brings a good summarization of previous works.

The conclusion section brings a summarization of the work done and ideas of future works related to the topic.

Author Response

Thanks for your review.

The current version has been reviewed by a native person.

Round 2

Reviewer 1 Report

The authors are presenting a review of the article that presents the PRISER model. Below I am presenting my comments:

1/ What is a taxonomic model?

2/ Could you present a comparison table in Section2, highlighting the gap in the area?

3/ How did you select the related work?

4/ What is the main section of the article? 3 or 4?  In other word, again, what is your main proposal?

5/  In Figure 5, do you have problems with scalability, since you have a single dispatcher?

6/ Why MQTT and not other, like COAP?

7/ Again, revisit the related work, how do you compare your system against initiates form 2019?

Author Response

First and foremost, we would like to thank the reviewer for his contribution, which has proven to

be very valuable in improving our paper. His comments have alerted us to several limitations

that we were previously unaware of and that we have since attempted to address.

Reviewer 2 Report

All my comments have been addressed in this revised manuscript. Just a minor comment, figures should always be self-explainable. So, in my opinion, fig. 1 should be revised abit to ease readability. for example, communication means what? from what i see below and in the text it is communication protocols. Please fix this minor comment.

Author Response

Firstly, we would like to thank the reviewer for his contribution, which proved to be invaluable in improving our article. The previous comments and this new assessment have alerted us to limitations that we did not previously know and which we have tried to address since then.

We made the necessary changes in Figure 01 and some more revisions and changes that can be observed throughout the text.

Thank you again.